# Ax: A Platform for Adaptive Experimentation

Miles Olson[1,*]  Elizabeth Santorella[1,*]  Louis C. Tiao[1,*]  Sait Cakmak[1,*]  Mia Garrard[1,*]
Samuel Daulton[1,*]  Zhiyuan Jerry Lin[1]  Sebastian Ament[1]  Bernard Beckerman[1]
Eric Onofrey[1]  Paschal Igusti[1]  Cristian Lara[1]  Benjamin Letham[1]  Cesar Cardoso[1]
Shiyun Sunny Shen[1]  Andy Chenyuan Lin[1]  Matthew Grange[1]  Elena Kashtelyan[1,*]
David Eriksson[1,*]  Maximilian Balandat[1,*]  Eytan Bakshy[1,*]

[1]Meta
[*]Equal contribution.

**Abstract**  Optimizing industry-scale machine learning systems involves resource-intensive black-box optimization. Adaptive experimentation substantially improves the sample efficiency of such tasks compared with naive baselines (such as grid or random search) by utilizing surrogate models and sequential optimization algorithms. Ax (https://ax.dev) is an open-source platform for adaptive experimentation. Ax is highly extensible and full-featured, and is used at scale at Meta. We discuss Ax's design, usage, and performance. Off the shelf, Ax achieves state-of-the-art performance in a wide range of synthetic and real-world black-box optimization tasks in machine learning, engineering, and science.

## 1 Introduction

Optimizing machine learning systems is a prominent application of black-box optimization. Modern machine learning pipelines involve multiple resource-intensive stages, including feature selection, architecture search, hyperparameter optimization, and optimization for inference and serving efficiency. Ultimately, deployed models integrate into higher-level systems, such as recommender systems, which require online A/B testing.

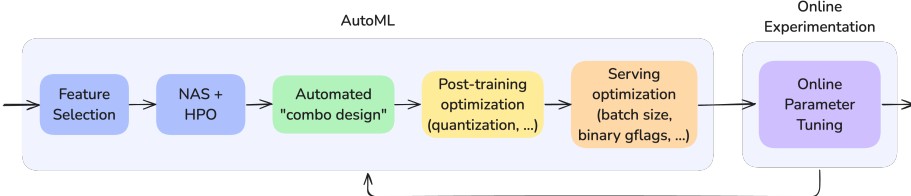

Figure 1: Industry-scale ML pipelines include diverse optimization tasks.

While each stage has unique design parameters and goals, many of these tasks can be formulated as *black-box optimization* problems. In such problems, the aim is to solve $\operatorname{argmax}_{x \in X} f(x)$ for some objective $f : X \to \mathbb{R}$, where evaluations $f(x)$ are the only available information and finding the optimum requires conducting *trials* that sample values $x \in X$.

When evaluations are costly, as they often are in modern ML pipelines, an iterative approach that adaptively explores the design space offers superior efficiency compared to naïve methods such as grid or random search (Turner et al., 2021). The performance of sample-efficient methods can be sensitive to implementation details, and a production optimization system must be robust to many real-world issues – for example, trial failures – while remaining accessible to people with diverse backgrounds, from those with little ML experience to people who implement their own optimization algorithms. To this end, we introduce Ax (https://ax.dev), a versatile, open-source,

adaptive experimentation platform implemented in Python for state-of-the-art sample-efficient optimization with a focus on accelerating the research-to-production pipeline.

## 1.1 Contributions

Ax supports automated, sample-efficient optimization for a variety of problems, including those frequently encountered in AutoML. Ax is feature-rich:

- Ax's expressive API handles complex search spaces, multiple objectives, constraints on both parameters and outcomes, and noisy observations, with or without observed noise levels. It supports suggesting multiple designs to evaluate in parallel (both synchronously and asynchronously) and can stop evaluations early to save resources.
- Ax provides sensible defaults, facilitating access to advanced techniques that are typically reserved for experts.
- Ax leverages state-of-the-art Bayesian Optimization (BO) algorithms implemented in BoTorch (Balandat et al., 2020) to deliver strong performance in a variety of problem classes.
- Ax allows researchers to customize optimization algorithms, models, and experimentation flows.
- Ax is production-ready, offering automation and orchestration features, as well as robust error handling for real-world deployment at scale.

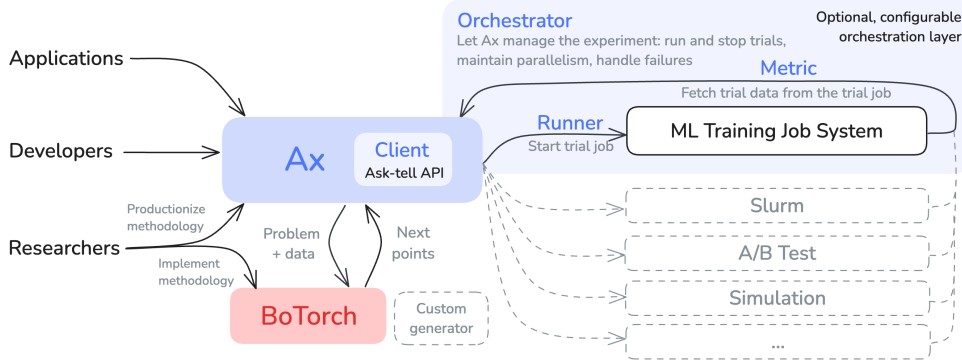

Figure 2: Illustration of Ax as a platform for adaptive experimentation

## 2 Related work

Several open-source packages exist for black-box / hyperparameter optimization (HPO), active learning, experiment management, and/or orchestration (i.e., handling trial execution, data fetching, and managing parallelism). SMAC (Lindauer et al., 2022), Nevergrad (Rapin and Teytaud, 2018), and Dragonfly (Kandasamy et al., 2020) offer black-box optimizers but not orchestration capabilities. Ray Tune (Liaw et al., 2018) integrates popular black-box optimizers including Ax, Optuna (Akiba et al., 2019), Nevergrad, HyperOpt (Bergstra et al., 2013), and BayesOpt (Martinez-Cantin, 2014) but only works with the Ray distributed ML platform. The Hydra (Yadan, 2019) configuration framework is commonly used in distributed computation and, like Ray Tune, supports various optimizers (Ax, Optuna, and Nevergrad). SyneTune (Salinas et al., 2022) implements various black-box optimizers and provides orchestration capabilities, primarily with Amazon Sagemaker. Vizier (Golovin et al., 2017a) exposes its optimization algorithm (Song et al., 2024) through an ask-tell service.

The features summary Table 1 shows that Ax provides a broader range of capabilities. In particular, imposing constraints on parameters and outcomes is not supported by most alternative, but is often needed in practice. Moreover, while other libraries focus exclusively on HPO, Ax also

supports A/B testing well: it handles substantial levels of (known) observation noise and offers batched trial representations to account for non-stationarity (Feng et al., 2025). Together, these features enable Ax to optimize the full ML pipeline in Figure 1.

| | Ax | Vizier | Syne Tune | Optuna | SMAC3 | HEBO |
|---|---|---|---|---|---|---|
| Single objective | ✓ | ✓ | ✓ | ✓ | ✓ | ✓ |
| Multiple objectives | ✓ | ✓ | ✓ | ✓ | ✓ | ✓ |
| Outcome constraints | ✓ | ✗ | ✓ | ✗ | ✗ | ✓ |
| Discrete/Mixed search spaces | ✓ | ✓ | ✓ | ✓ | ✓ | ✓ |
| Hierarchical search spaces | ✓ | ✓ | ✗ | ✓ | ✓ | ✗ |
| Parameter constraints | ✓ | ✗ | ✗ | ✗ | ✓ | ✗ |
| Time series observations | ✓ | ✓ | ✓ | ✓ | ✓ | ✗ |
| Noise measurements | ✓ | ✗ | ✗ | ✗ | ✗ | ✗ |
| Closed-loop orchestration | ✓ | ✗ | ✓ | ✓ | ✓ | ✗ |
| Early stopping | ✓ | ✓ | ✓ | ✓ | ✓ | ✗ |
| Visualization | ✓ | ✓ | ✓ | ✓ | ✗ | ✗ |

Table 1: Overview of supported features for popular open-source adaptive experimentation libraries.

**Ax and BoTorch.** BoTorch (Balandat et al., 2020) is a popular library for Bayesian optimization (BO) research built on PyTorch. Ax has a special relationship with BoTorch, illustrated in Figure 2; it leverages components implemented in BoTorch, and the two libraries are developed in tandem as sister projects. This allows for a separation of concerns, where BoTorch provides a modular and extensible interface for composing BO primitives, while Ax provides a higher-level interface and manages the experimentation process end-to-end. This design empowers researchers who use BoTorch to deploy new methods via Ax with minimal boilerplate, accelerating the research-to-production pipeline.

## 3  Usage of Ax

### 3.1  Open-Source

Ax is open-source (MIT license) and has an active developer community. Its Github repository `https://github.com/facebook/Ax` demonstrates a commitment to ongoing improvements to the package and provides a forum for teaching and discussion. At the time of writing, Ax has around 4,250 commits across 3,000 pull requests from 100 unique developers. More than 2,500 Github users have starred the repository, created more than 330 forks, and opened more than 800 issues.

### 3.2  Deployment at Meta

Ax has been deployed at scale at Meta to help solve some of the company's most challenging optimization problems. Its primary use cases at Meta are offline optimization of ML hyperparameters, parameter tuning of ML systems with online experiments, infrastructure optimization, and hardware design. Ax is also used independently by researchers and engineers across Meta to solve various ad hoc problems.

Ax emphasizes quality, stability, and robustness. It is fully typed and type-checked in Python 3.10+, performs extensive run-time validation, its unit tests cover >96% of its >52,000 lines of code, and its integration tests and nightly benchmarks ensure API and performance stability over time. Ax maintains high levels of reliability in production services at Meta.

**Hyperparameter optimization.** AxSweep, a deployment of Ax on Meta's internal ML infrastructure, is used across the company to optimize hyperparameters of ML systems, including learning rates, architecture parameters, training data weights, training and serving configurations, and general ML infrastructure parameters. In 2024, more than a thousand engineers and scientists across the company ran more than 70,000 parameter tuning experiments through AxSweep. In addition, some teams have built specialized tools using custom interfaces and orchestration logic while leveraging Ax as the core optimization engine.

**Online Parameter Tuning.** Often, parameter changes can only be reliably evaluated by testing the changes in a large-scale online experiment, otherwise known as an *A/B test*. Ax is integrated with Meta's A/B testing systems and is used across product groups for various online use cases, including optimizing ranking and retrieval configurations, tuning infrastructure parameters for capacity efficiency, and optimizing on-device content retrieval policies (Letham and Bakshy, 2019). In 2024, more than 200 different users ran more than 1,000 Ax experiments for online tuning.

Ax can handle particular challenges of the online setting, including delayed feedback due to long-term effects, non-stationarity of the effect of parameter changes over time, highly noisy observations, and contextual policy optimization (Feng et al., 2020, 2025). It supports batch-sequential optimization, in which each arm within a batch is compared against a common control arm (however, as of this writing, this is not yet exposed in the user-friendly `Client` API).

**Hardware Design & Simulation Optimization.** Due to the resources required to run computer simulations or to manufacture and test prototypes in the lab, designing novel hardware for ML (such as AI training/inference accelerators) and Augmented/Virtual Reality (AR/VR) also involves solving challenging black-box optimization problems. At Meta, Ax has been used extensively for these purposes, including optimizing the design of the waveguides (optical nano-structures) in the development of Meta's Orion AR glasses (Meta, 2024a), which required supporting a high-throughput optimization setting with tens of thousands of evaluations and the development of novel high-dimensional multi-objective optimization algorithms, e.g., Daulton et al. (2022).

## 4 API and Usage Patterns

Ax defines a concise yet expressive API. A typical optimization involves the user:

1. **Configuring the experiment**, including the search space, optimization goals and constraints
2. **Conducting the experiment**, either in an *ask-tell* fashion where candidates are manually requested and data is manually reported, or in a *closed-loop* fashion where trials are run automatically via `Metric` and `Runner` abstractions.
3. **Analyzing the experiment** through provided diagnostic visualizations and tables.

This process is facilitated by Ax's `Client` class, which serves as a single entry point, exposing methods for each task and managing experiment state. Ax supports saving experiment state to `json` and remote databases (MySQL, PostgreSQL).

### 4.1 Configuring the experiment

Ax uses config classes, lightweight containers which group related configuration settings together and validate the setup at instantiation time. This improves API clarity and provides a serializable interface suitable for deployments where Ax is called over-the-wire. After instantiating a `Client` object, configuring an Ax optimization consists of three steps as illustrated in Code Example 1.

First, the user calls `configure_experiment()` to define the search space and set other metadata useful for managing experiments such as a name, description, or owner. Search spaces are defined by a collection of parameter configs: A `RangeParameterConfig` describes continuous design parameters with configurable bounds, scaling (linear or logarithmic), and optionally step size. A

```
from ax import *
client = Client()

client.configure_experiment(
    parameters=[
        RangeParameterConfig(name="n_layers", bounds=(1, 16), parameter_type="int"),
        RangeParameterConfig(name="learning_rate", bounds=(1e-8, 1), parameter_type="float", scaling="log"),
        ChoiceParameterConfig(name="batch_size", values=[2,4,8,16,32,64], parameter_type="int", is_ordered=True),
    ],
    parameter_constraints=[...],
)
client.configure_optimization(objective="accuracy", outcome_constraints="model_size_MB <= 2_500")
```

Code Example 1: Experiment Configuration in Ax for minimizing the loss subject to a constraint on the model size not exceeding 2.5GB.

ChoiceParameterConfig describes a discrete parameter, either ordinal or categorical. A user may also constrain the search space via a collection of linear inequalities, or specify dependents on choice parameters to define a hierarchical structure.

Next, the user calls configure_optimization() to specify objective(s) and optionally outcome constraints to define optimization goals and guardrails. Optionally, a user may call configure_generation_strategy() to control aspects of the optimization process such as the initialization budget or selecting between preset optimization methods.

## 4.2 Conducting the experiment

An experiment can be conducted either in an *ask-tell* fashion where candidates are manually requested and data are manually reported, or in a *closed-loop* fashion where trials are run automatically via previously configured Metric and Runner abstractions.

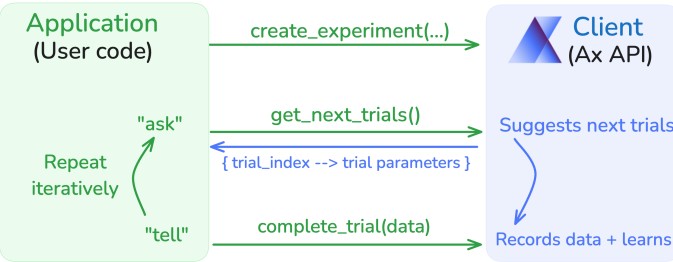

Figure 3: Ax API ('Client') in "ask-tell" mode

**Ask-tell experimentation**. Running an experiment in *ask-tell* mode (Figure 3, Code Example 2a) can be useful if trials are executed in the same Python runtime as Ax or if trials require manual intervention to deploy and evaluate. In this setting, users request one or more candidate parameterizations at a time using get_next_trials(), evaluate the candidate(s) externally to Ax, and report the results back via complete_trial(). Evaluations can be attached once or multiple times using a progression term to indicate a timeseries-like structure as found in learning curves. Users may also mark trials' status or query whether the trial should be terminated early to preserve experimentation budget.

**Closed-loop experimentation**. Alternatively, running an experiment in *closed-loop* mode (Figure 4, Code Example 2b) is useful when trials are executed on external systems or when it is desirable to fully automate the experimentation loop. Users must define a Runner class which implements logic

```
for _ in range(num_trials):
    trials = client.get_next_trials(max_trials=1)
    for i, params in trials.items():
        # User-defined function / external process
        loss = train_and_evaluate(**params)
        client.complete_trial(
            trial_index=i,
            raw_data={"loss": loss},
        )
```

```
client.configure_runner(runner=Runner(...))
client.configure_metrics(metrics=[Metric(...)])

client.run_trials(  # Runs all trials automatically
    max_trials=30,
    parallelism=4,
    tolerated_trial_failure_rate=0.1,
    initial_seconds_between_polls=1,
)
```

(a) Ask-tell operation (fully sequential)      (b) Closed-loop operation

Code Example 2: Running experiments with Ax's `Client` API.

for deploying trials to external systems and polling their status and a `Metric` class for fetching a trial's results. For instance, this could entail writing code for enqueuing an ML workload on a HPC cluster using SLURM and reading the associated learning curves from Tensorboard. Then, a user simply makes calls `run_trials()`, specifying a maximum number of trials to run and miscellaneous options to be consumed by Ax's `Orchestrator`, the finite state machine responsible for managing the optimization loop. The `Orchestrator` then runs the experiment automatically, supporting asynchronous parallelism (including early stopping), and providing robust error handling, including the ability to resume interrupted optimizations.

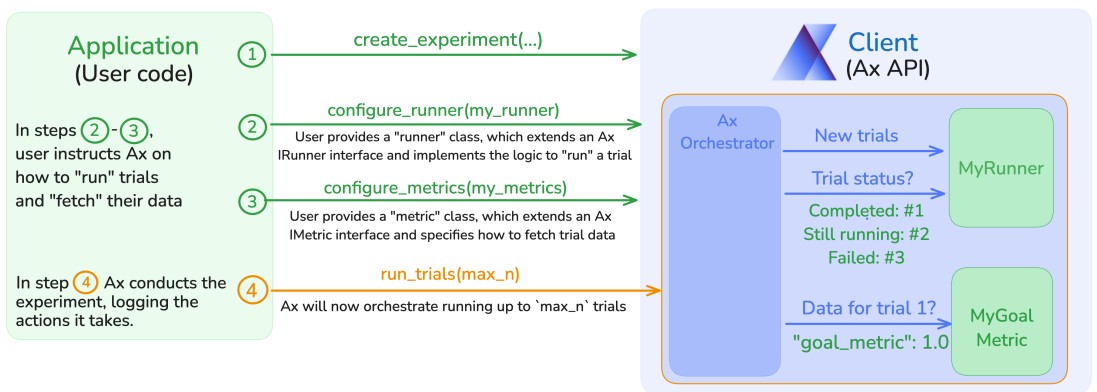

Figure 4: Ax API ('Client') in "closed-loop" mode

### 4.3 Analyzing the experiment

During or after experimentation, users may extract the best point (or the Pareto frontier in multi-objective settings) from Ax, or may use Ax's suite of analysis tools to understand their results and gain deeper insights into the underlying black-box problem. Ax provides a rich framework for generating plots and tables via its `Analysis` class. At any point during experimentation, users can generate analyses individually, implement their own custom analyses, or allow Ax to heuristically select the most relevant analyses for their specific setting. These may include scatter plots of outcomes, global sensitivity analyses (Sobol', 2001), and cross-validation, slice, and contour plots based on the underlying surrogate model. See Figure 5 for two examples provided out-of-the-box by Ax, and Appendix D for additional details.

## 5 Candidate Generation

Selecting the right strategy to adaptively generate the next point to evaluate is a non-trivial task, and the optimal choice will depend on the characteristics of the problem. By default, Ax uses

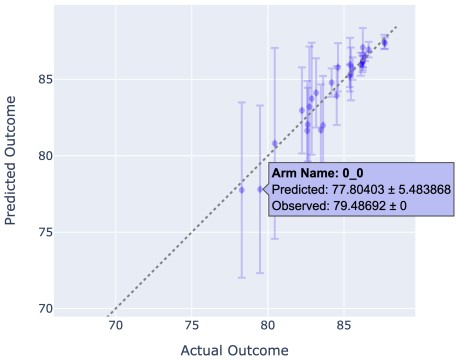

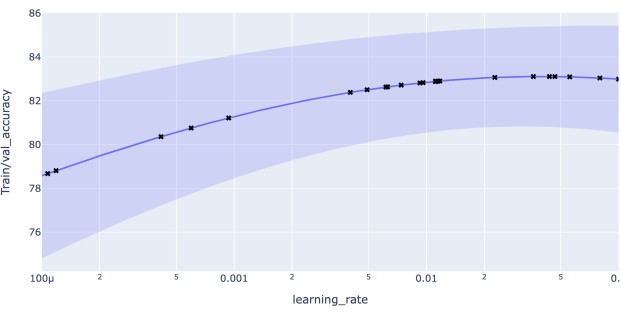

(a) Leave-one-out cross-validation

(b) 1D Slice plot of model predictions as (other parameters fixed at reference)

Figure 5: Example visualizations provided by Ax that help understand surrogate model quality and behavior of the underlying black-box function. See Appendix D for additional details.

heuristics to dispatch to various BO algorithms implemented in BoTorch (see Section 5.2 for details) which, as our benchmarks in Section 6 demonstrate, achieve good performance out-of-the box across a broad range of use cases.

## 5.1 The `GenerationStrategy` abstraction

Ax's GenerationStrategy is a flexible abstraction that dynamically transitions between sampling / point suggestion methods, allowing different algorithms to be used at different stages. The GenerationStrategy comprises GenerationNodes, which suggest candidates to evaluate, and TransitionCritera, which determine when and how to switch between nodes. These form a finite-state machine which is capable of encoding complex optimization procedures. Users can define their own optimization algorithms by defining GenerationNodes; in fact, all baselines in this paper were evaluated with Ax itself, wrapping each external method in a GenerationNode.

## 5.2 Default model dispatch and typical settings

By default, Ax automatically constructs a GenerationStrategy using the search space of the problem, optimization goals, and user settings. Ax evaluates the center of the search space, runs four quasi-random Sobol trials, then transitions to Bayesian optimization for the rest of the experiment.

For Bayesian optimization, Ax defines a *Modular BoTorch framework*, which simplifies using probabilistic surrogates (typically Gaussian process models) and acquisition functions implemented in BoTorch within a GenerationNode. As many surrogate models are defined over continuous/numerical domains, a series of transforms are applied within the Adapter layer. This flattens potential hierarchical parameter structure, ensures all parameters are numerical, and applies any necessary scaling (e.g. log-scaling) specified by the search space or optimization settings.

The low-level modeling and candidate generation is then handled by Ax's *Modular BoTorch Generator* (MBG) interface. Besides exposing BoTorch surrogate models and acquisition functions, this interface also provides advanced capabilities such as per-metric model selection between multiple models based on a measure of model fit quality. By default, MBG will pick an appropriate surrogate class (typically a SingleTaskGP with an RBF Kernel and dimension-scaled priors (Hvarfner et al., 2024)), an acquisition function (qLogNEI for single- or qLogNEHVI for multi-objective optimization (Ament et al., 2023)), and an optimizer suitable for the problem search space (L-BFGS-B for fully continuous spaces, enumeration for small discrete spaces, alternating between discrete and continuous steps for mixed spaces). For additional details, see Appendix C.1.

## 5.3 Early stopping

For use cases where partial results on the outcomes are available while trials are running (such as learning curves in ML model training), Ax implements an `EarlyStoppingStrategy` interface. This interface integrates with the `Client` API and allows easy extension of Ax with custom strategies for early-stopping / pruning trials. In ask-tell mode, the user queries the `Client`'s `should_stop_trial_early()` method for which trials to stop manually. In closed-loop mode, the stopping strategy is automatically queried by the `Orchestrator`, and to-be-pruned trials are stopped automatically via the `Runner`. Ax ships with a robust, model-free default `PercentileEarlyStoppingStrategy` generalizing the Median Stopping Rule (Golovin et al., 2017b) that prunes trials based on their performance relative to other trials at the same progression.

## 6 Benchmarks

Ax provides a flexible benchmarking setup that allows easily evaluating different algorithms—including external optimizers with ask-tell interfaces—while using Ax as the orchestration layer. Here, we use this setup to benchmark Ax against a number of popular black-box optimization libraries with an AutoML focus.

### 6.1 Experimental Setup

**Baselines**. We compare Ax to Vizier (Golovin et al., 2017a; Song et al., 2022, 2024), Optuna (Akiba et al., 2019), SMAC3 (Lindauer et al., 2022), HEBO (Cowen-Rivers et al., 2022), and random search. We use the default configurations for Vizier, Optuna, HEBO, and Ax. For SMAC3, we consider both the BlackBoxFacade (Gaussian process) and HyperParameterOptimizationFacade (Random forest) implementations. We ran 10 replications for each of the slower methods (Ax, HEBO, SMAC-BB, and Vizier) and 100 for the faster methods (Optuna, random search, and SMAC-HPO).

We compare Ax's default early-stopping functionality to several state-of-the-art early-stopping methods implemented in Optuna: ASHA (Li et al., 2020), Hyperband (Li et al., 2018), BOHB (Falkner et al., 2018), and the Median Stopping Rule (Golovin et al., 2017b). All methods are run with asynchronous parallelism of 4. We ran 20 replications for all early-stopping experiments. Details on the specific Optuna configurations are provided in Appendix A.1.

**Problems**. We consider a comprehensive suite of 47 popular benchmark problems from the literature covering various settings, including single- and multi-objective optimization, black-box outcome constraints, noisy observations, continuous, discrete, and mixed search spaces, sequential and parallel evaluation of trials, and problems with progressive evaluations that enable early-stopping decisions based on intermediate performance. Table 3 in the Appendix contains the full list of problems and additional details.

**Benchmark scoring**. We leverage a scoring system similar to Turner et al. (2021) to aggregate results across different problems. Assuming the goal is to minimize a single objective $f(x)$, we define the score for a given replication after $t$ iterations on problem $p$ as:

$$100 \times \left(1 - \frac{f_p^* - \min\{f_1, \ldots f_t\}}{f_p^* - \mathrm{RS}_{5p}}\right), \tag{1}$$

where $f_p^*$ is the global optimal value and $\mathrm{RS}_{5p}$ is the best value observed after 5 trials with Ax's random search (including starting at the center of the search space), averaged over 100 replications. $f_i$ is the (noise-free) value of the $i$-th parameterization. So an optimal solution gets a score of 100, and a method that matches the average performance of 5 trials of random search gets a score of 0. This scoring method extends to the multi-objective setting by using the hypervolume of the Pareto frontier in place of the best observation of $f$. For the constrained setting, we follow the rationale

in Hernández-Lobato et al. (2016) and subsequent work that any feasible solution is better than an infeasible solution and score infeasible solutions using the worst-seen feasible objective value (across all benchmarks for the problem).

For early-stopping, where methods utilize computational resources adaptively, we depart from this normalized scoring and instead report the best observed objective value as a function of total training epochs consumed, allowing direct comparison of resource efficiency across methods.

For noisy problems, scoring uses true underlying function values rather than observed values; this is equivalent to omnisciently selecting the best in-sample point (or points, if multi-objective) and evaluating those parameters noiselessly. This measures the quality of the generated arms in isolation from the ability of the package to select the best arm. This approach has been taken in previous works, which highlighted that results were generally consistent with selecting the best in-sample point using the surrogate model (Daulton et al., 2021).

## 6.2 Results

The results for the different benchmark problems aggregated across 8 different categories are shown in Table 2 and Figure 6, with additional results for early-stopping in Figure 7. We observe that Ax is competitive with all baselines across all problem settings and outperforms baselines substantially on mixed/discrete, multi-objective, constrained, and noisy problems. This is largely attributed to its use of state-of-the-art algorithms such as BoTorch's qLogNoisyE(HV)I and discrete/mixed acquisition function optimizers that can effectively handle binary, discrete, and continuous parameters. Additional details, including anytime performance and runtimes, are provided in Appendix A.3.

| Category Method | BBOB | Other vanilla | Mixed/Discrete | Async | Multi-obj | Noisy | Constr. | High-Dim |
|---|---|---|---|---|---|---|---|---|
| Ax | 73.9 | 98.8 | 69.4 | 78.1 | 90.1 | 80.0 | 97.2 | 70.1 |
| Vizier | 74.1 | 99.1 | 55.9** | 74.2 | 65.4** | 61.3** | | 53.7** |
| SMAC-BB | 79.1** | 98.1 | 23.8** | 99.1 | 66.0** | 72.8** | | 40.2** |
| HEBO | 81.7** | 99.4 | 50.7** | 85.7 | 64.8** | 61.5** | 70.4** | 33.7** |
| Optuna | 35.8** | 82.8** | 51.4** | 76.3** | 48.6** | 47.2** | | 34.1** |
| SMAC-HPO | 41.9** | 71.5** | 15.2** | 85.9** | 46.7** | 43.2** | | 27.6** |
| Random Search | 17.2** | 67.1** | 46.5** | 71.6** | 35.5** | 35.0** | 38.8** | 22.0** |

Table 2: Performance matrix across libraries and benchmark problems in terms of score (1). Here * and ** denote a statistically significant difference from Ax at the 95% and 99% level, respectively.

## 7 Future work

At the time of writing, the features in Table 1 are fully supported in Ax's top-level `Client` API and comprehensively tested and documented. Ax has additional capabilities that are not yet API-stable, including cost-aware multi-fidelity optimization, transfer learning (leveraging data from related experiments), active learning, and preference-based optimization (e.g., based off of pairwise comparisons between outputs of a generative model, as illustrated in Appendix E.1). We are working towards improving the support and usability of these features going forward.

A current limitation of Ax is that it was initially designed for small-sample regimes and, due to internal overhead does not efficiently handle the thousands of trials per experiment required by high-throughput methods such as TuRBO (Eriksson et al., 2019). We are working to reduce this overhead.

## 8 Broader Impact Statement

Ax is a tool for general black-box optimization and, as such, does not present any specific risks.

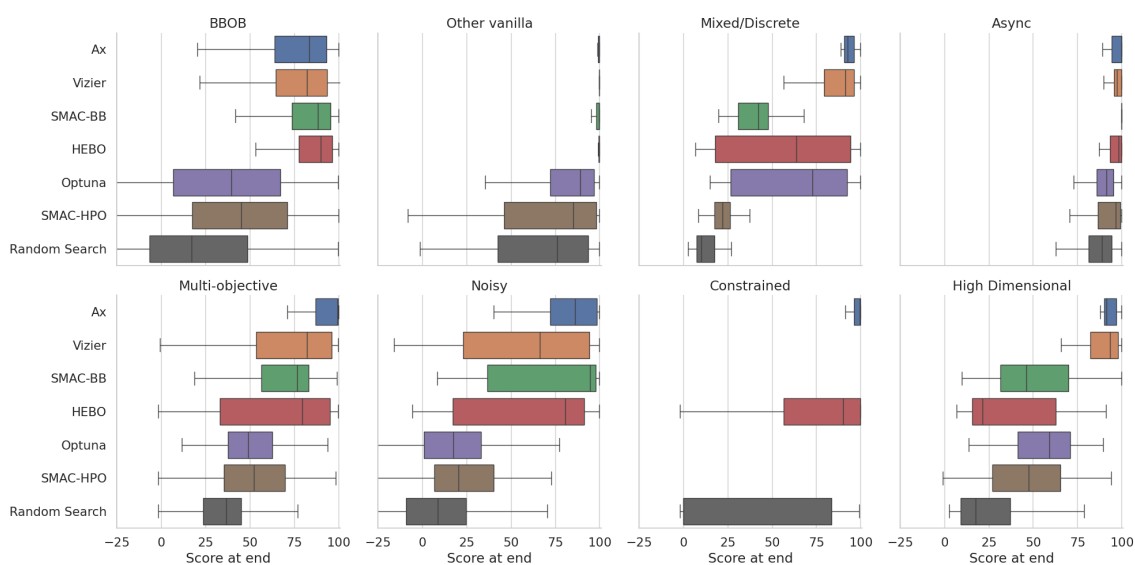

Figure 6: Final performance of each method aggregated across 8 different problem types.

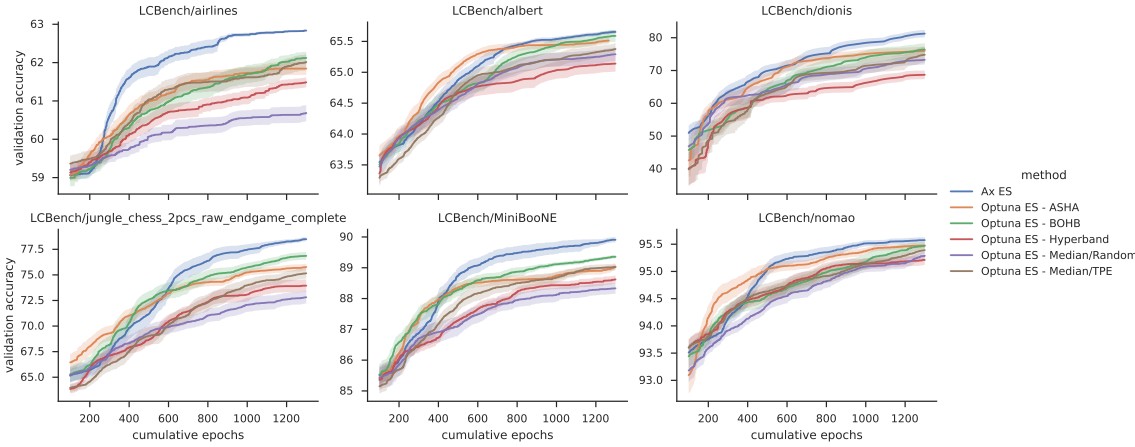

Figure 7: Best validation accuracy achieved by each early-stopping method as a function of cumulative epochs across six randomly selected LCBench learning curve benchmark datasets.

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

## A  Details on the Experiments

In this section, we provide some additional details on the benchmarks.

### A.1  Baselines

**Baseline Licenses**. Vizier uses an Apache 2.0 license[1], SMAC3 uses a BSD-3 license[2], HEBO uses an MIT license[3], Optuna uses an MIT license[4].

**Baseline Algorithms**. Vizier at its core uses a Bayesian optimization algorithm based on GP models similar to that of Ax (Song et al., 2024). The main differences are that (i) it uses an Upper Confidence Bound (UCB) acquisition function in conjunction with a trust-region algorithm for single-objective optimization and (ii) a version of the Firefly genetic algorithm for acquisition function optimization.

HEBO (Cowen-Rivers et al., 2022) also uses a GP model and performs outcome transformations and input warping to better handle heteroskedasticity and non-stationarity. It uses a compound acquisition function that employs a generic algorithm to solve a multi-objective optimization problem over the values of multiple classic (Expected Improvement, Upper Confidence Bound, and Probability of Improvement) acquisition functions to generate new candidates.

SMAC3 implements multiple algorithms; in this comparison we consider SMAC-BB and SMAC-HPO. Both algorithms default to a quasi-random initial design. SMAC-BB implements a classic GP-based Bayesian optimization algorithm, using a Gaussian Process surrogate with Matérn 5/2 kernel and Expected Improvement as the acquisition function. SMAC-HPO uses a Random Forest surrogate with logEI acquisition function.

Optuna by default uses an approach that, rather than fitting a regression model to the (potentially transformed) outcomes, models a density ratio via Tree-Structured Parzen (TPE) estimators (Bergstra et al., 2011) is used to express an approximation of the the Expected Improvement acquisition function directly (up to a constant factor). The main benefit of this approach is that it can be very fast. Additionally, Optuna implements early-stopping functionality through a modular system of pruners and samplers that can be combined to realize various methods. The pruners we evaluate include:

- `MedianPruner`: Implements the median-rule stopping rule (Golovin et al., 2017b)
- `SuccessiveHalvingPruner`: Implements the Successive Halving mechanism
- `HyperbandPruner`: Implements the Hyperband mechanism

These pruners are paired with the `RandomSampler` and `TPESampler` samplers. The specific pruner-sampler combinations correspond to different early-stopping methods:

- `SuccessiveHalvingPruner` + `RandomSampler` → ASHA (Li et al., 2020)
- `HyperbandPruner` + `RandomSampler` → Hyperband (Li et al., 2018)
- `HyperbandPruner` + `TPESampler` → BOHB (Falkner et al., 2018)
- `MedianPruner` + `RandomSampler` → Median-rule with random search
- `MedianPruner` + `TPESampler` → Median-rule with TPE-based Bayesian optimization

### A.2  Benchmark Problem Details

A summary of the benchmark problems is provided in Table 3. All test problems considered in this paper are publicly available. Similar to the Google Vizier paper (Song et al., 2024), we consider randomly shifted versions of the 24 single-objective BBOB problems (Elhara et al., 2019). In the

---

[1] `https://github.com/google/vizier/blob/main/LICENSE`
[2] `https://github.com/automl/SMAC3/blob/main/LICENSE.txt`
[3] `https://github.com/huawei-noah/HEBO/blob/master/HEBO/LICENSE`
[4] `https://github.com/optuna/optuna/blob/master/LICENSE`

multi-objective setting, we consider the car side impact and vehicle safety problems from Tanabe and Ishibuchi (2020), the penicillin problem (Liang and Lai, 2021), and the popular synthetic ZDT1-3 and DTLZ2 problems.

For the constrained setting, we use the tension compression string, pressure vessel, and welded beam problems that were considered in Eriksson and Poloczek (2021). The LABS (50 binary parameters) and mixed Ackley (50 binary parameters, 3 continuous parameters) problems from Deshwal et al. (2023) tests how well different methods can deal with binary and mixed search spaces.

NASBench (Ying et al., 2019) provides fully tabulated results of different convolutional NN architectures trained and evaluated on the CIFAR-10 data set. NASBench uses an Apache 2.0 license.

The LCBench benchmark (Zimmer et al., 2021) provides learning curve observations for various performance metrics collected during the training of fully-connected neural networks (NN). These networks were trained for 50 epochs across 2,000 randomly sampled hyperparameter configurations and evaluated on 35 datasets from the AutoML Benchmark (Gijsbers et al., 2019) hosted on OpenML (Vanschoren et al., 2014). The LCBench benchmark data is available under the Apache 2.0 license, while the OpenML datasets are distributed under the CC BY 4.0 license. The hyperparameter search space consists of 7 parameters: 3 integer-valued and 4 float-valued. Since LCBench cannot possibly provide exhaustive evaluations of the search space, we use a multi-output random forest surrogate model–where each output corresponds to a different epoch–to interpolate evaluations between observed configurations.

### A.3 Additional Benchmark Results

Figure 8 averages the time to run all the trials in an optimization within each problem group. We observe an expected bifurcation in run times according to the optimization approach: Optuna and SMAC-HPO, which do not use Gaussian process models, run substantially faster than Ax, Vizier, and SMAC-BB, which do use GPs. Optuna, which by default uses a likelihood-free Bayesian Optimization approach based on Tree Parzen Estimators (TPEs), has a particularly fast runtime. However, as the results in Section 6.2 demonstrate, this comes at a substantial loss in optimization performance across all problem types.

Within the GP-based libraries, we see that Ax's run time generally falls between that of HEBO (faster) and Vizier and SMAC-BB (slower), except for the multi-objective problems, where Ax is the slowest method. However, this time investment pays off in terms of optimization performance, where Ax far outperforms all other baselines. Note also that this slowless is largely because the hypervolume computations were – in order to enable a fair comparison – not being run on a GPU, which would lead to substantially speed ups, see Daulton et al. (2020).

Figure 9 illustrates the anytime performance of the different libraries across the different problem settings. We observe that in the multi-objective, constrained, and high-dimensional settings, Ax achieves substantially better performance early on. The situation is a bit more nuanced in the mixed/discrete setting, where Ax goes head-to-head with most other libraries until about 20 trials, after which it starts outperforming the other ones. We conjecture that one of the main reasons for this is the superior acquisition function optimization algorithm in Ax for discrete and mixed spaces that do not rely on continuous relaxation or purely discrete optimizers.

### A.4 Resources used for benchmarking

To enable a fair comparison, all evaluations were run on the same type of virtual machine with 32 x86 cores (physical CPU: Intel Xeon) and 48GB of memory. In total, the benchmarks in this paper required around 550 machine-hours of compute – as the benchmark problems were either synthetic or based on tabular data or surrogates, the additional cost from evaluating the underlying black-box function is negligible.

| Problem setting | Test Problems | Dim. | Objectives | Constraints | Evaluation budget |
|---|---|---|---|---|---|
| Single-objective | BBOB1-24 | 20 | 1 | 0 | 50 trials |
| Other | Branin | 2 | 1 | 0 | 50 trials |
| vanilla | SixHumpCamel | 2 | 1 | 0 | 50 trials |
| single-objective | Hartmann | 6 | 1 | 0 | 50 trials |
| Constrained | Tension Compression String | 3 | 1 | 4 | 50 trials |
| | Pressure Vessel | 4 | 1 | 4 | 50 trials |
| | Welded Beam | 4 | 1 | 5 | 50 trials |
| Multi-objective | Car Side Impact | 7 | 4 | 0 | 50 trials |
| | ZDT1-3 | 5 | 2 | 0 | 50 trials |
| | DTLZ2 | 6 | 2 | 0 | 50 trials |
| | Penicillin | 7 | 3 | 0 | 50 trials |
| | Vehicle Safety | 5 | 3 | 0 | 50 trials |
| Mixed/Discrete | LABS | 50 | 1 | 0 | 100 trials |
| | Mixed Ackley | 53 | 1 | 0 | 100 trials |
| | NASBench201/ImageNet16-120 | 6 | 1 | 0 | 50 trials |
| | NASBench201/CIFAR-100 | 6 | 1 | 0 | 50 trials |
| High-dimensional | Embedded Hartmann | 30 | 1 | 0 | 50 trials |
| | LABS | 50 | 1 | 0 | 100 trials |
| | Mixed Ackley | 53 | 1 | 0 | 100 trials |
| Early-stopping/Async | LCBench/airlines | 7 | 1 | 0 | 1,400 epochs |
| | LCBench/albert | 7 | 1 | 0 | 1,400 epochs |
| | LCBench/dionis | 7 | 1 | 0 | 1,400 epochs |
| | LCBench/jungle_chess_... | 7 | 1 | 0 | 1,400 epochs |
| | LCBench/MiniBooNE | 7 | 1 | 0 | 1,400 epochs |
| | LCBench/nomao | 7 | 1 | 0 | 1,400 epochs |
| Async | Branin | 2 | 1 | 0 | 50 trials |
| | NASBench201/ImageNet16-120 | 6 | 1 | 0 | 50 trials |
| | NASBench201/CIFAR-100 | 6 | 1 | 0 | 50 trials |
| Noisy | Branin | 2 | 1 | 0 | 50 trials |
| | BBOB01 | 20 | 1 | 0 | 50 trials |
| | DTLZ2 | 6 | 2 | 0 | 50 trials |
| | Hartmann | 6 | 1 | 0 | 50 trials |

Table 3: Summary of the benchmark problems used in the paper and how they are grouped for analysis.

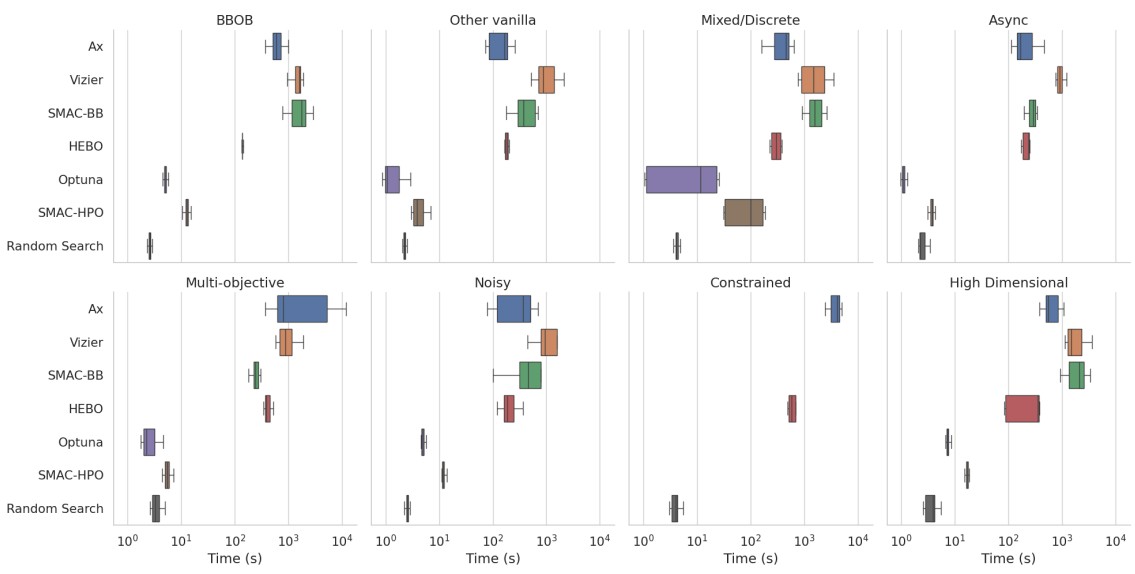

Figure 8: Runtime of each method aggregated across 8 different problem types (log scale).

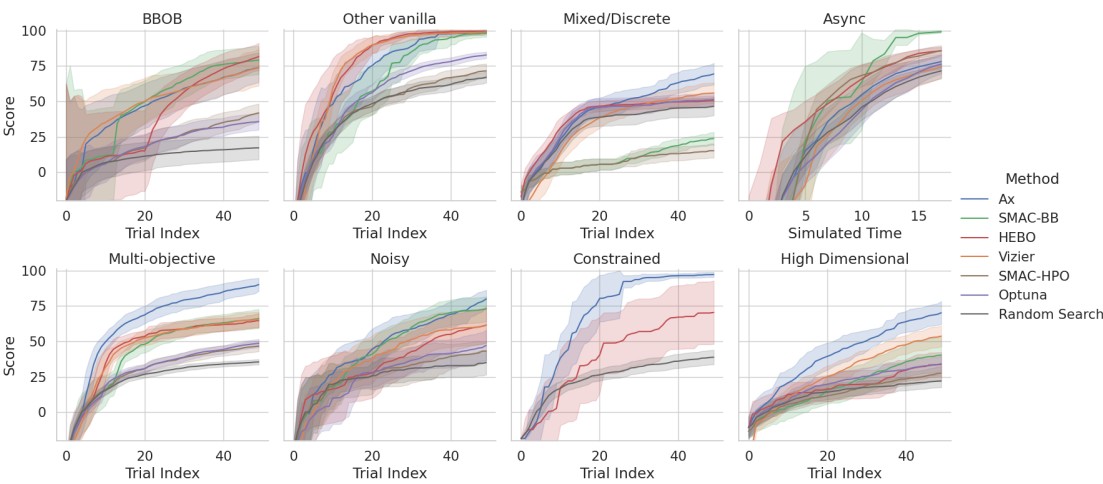

Figure 9: Score on a per-trial basis. Trajectories for problems that typically run more trials have been truncated so that all problems in the same grouping have the same trajectory length.

## B API Details

Users interact with Ax via its `Client`, which manages experiment state either through an ask-tell paradigm or with orchestrated trial deployment and data retrieval. Figures 10 and 11 provide additional details on the ask-tell and orchestrated operations of Ax.

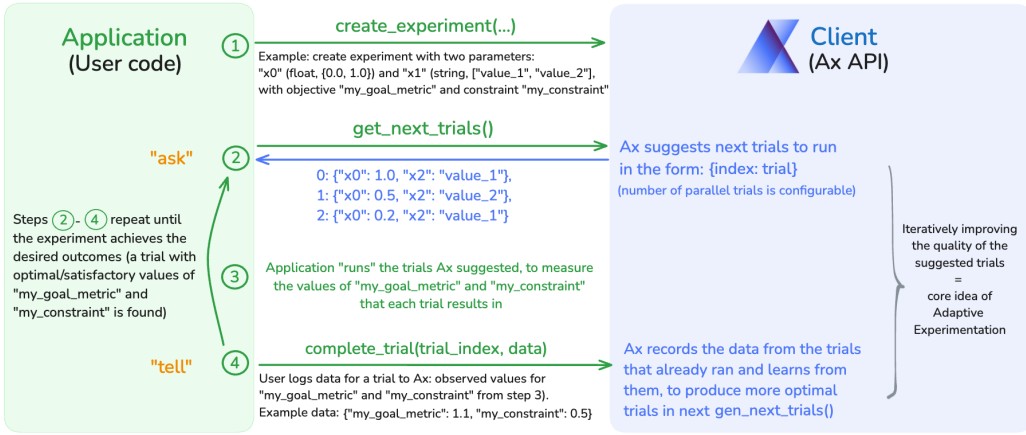

Figure 10: Details on the Ax API ('Client') in "ask-tell" mode

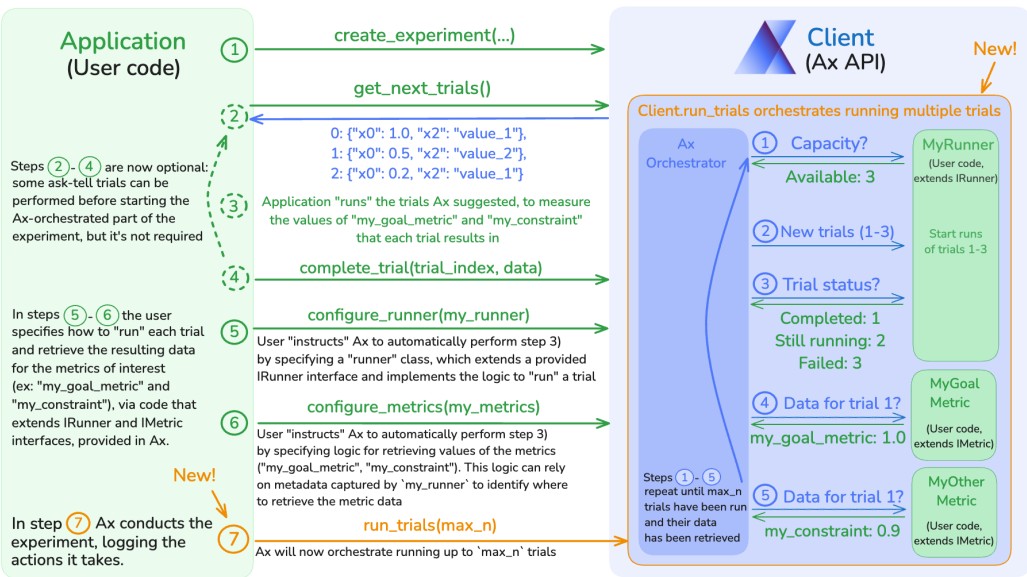

Figure 11: Details on the Ax API ('Client') in "orchestrated" mode

Advanced users and developers are able to control Ax's candidate generation process via modifications to its `GenerationStrategy` or by creating custom `GenerationNodes`, which can implement any optimziation strategy. Figure 12 illustrates a typical `GenerationStrategy` comprised of three `GenerationNodes` representative of the default optimization strategy used in Ax including for generating the benchmark results in Section 6. Figure 13 shows a more complex strategy that is useful in contexts where certain advanced BO methods such as transfer learning can become applicable partway through the experimentation process.

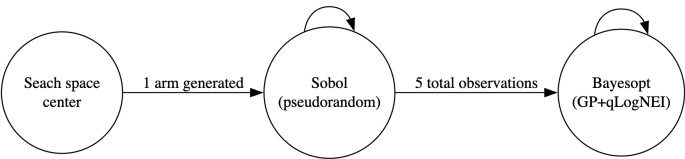

Figure 12: A simple `GenerationStrategy` with two initialization nodes followed by a BO node

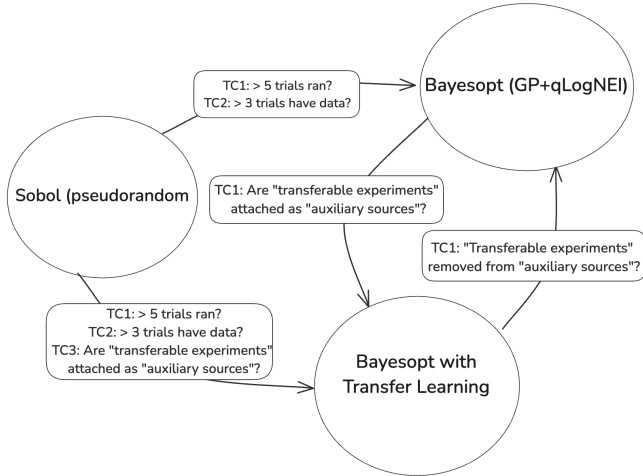

Figure 13: A `GenerationStrategy` which utilizes BO with Transfer Learning when available and appropriate

## C Details on Candidate Generation in Ax

### C.1 Configuration of Modular BoTorch Generator

*Modular BoTorch Generator* (MBG) offers a convenient interface for leveraging BoTorch models and acquisition functions for candidate generation in Ax. MBG consists of two core components: `Surrogate` and `Acquisition`. Before going into details on these components, it is worth noting that the capabilities and default behavior of MBG are regularly updated to support new use cases and incorporate new methods and developments to improve the out-of-the-box performance of Ax.

**Surrogate**. The `Surrogate` handles all modeling-related functionality, including construction and fitting of BoTorch models, and advanced features like selecting between multiple fitted BoTorch models based on a measure of model quality (under active development), such as the marginal log-likelihood or rank correlation computed on cross-validation outcomes. The `Surrogate` will select an appropriate BoTorch model class to use, based on the features of the search space (after applying transforms in `Adapter`) such as `MultiTaskGP` in the presence of task features, `SingleTaskMultiFidelityGP` in the presence of fidelity features, and `SingleTaskGP` for most other use-cases.

The `Surrogate` can be customized extensively using `SurrogateSpec` and `ModelConfig`. In addition to specifying an off the shelf BoTorch model class to use, various options including the covariance and likelihood modules, input and outcome transforms, and the marginal log-likelood used for model fitting can be customized. `SurrogateSpec` can include multiple `ModelConfigs`, in which case multiple models are fit and the best one for each metric (selected using a specified evaluation criterion on the cross-validation outcomes) will be used within the acquisition function. Code Example 3 shows an example with two `ModelConfigs`.

**Transforms**. Ax comes with a comprehensive transform layer that is used to transform the trials and optimization config from the user-specified search space into a "modeling space" that is more appropriate for the underlying optimization algorithms. This includes input transforms such as `OrderedChoiceToIntegerRange` which converts ordered choice parameters into a contiguous range $0, 1, ..., n_{\text{choices}} - 1$ of integers, `OneHot` which one-hot encodes unordered choice parameters, `Log` which transforms parameters in log-scale, and `Normalize`/`UnitX` which normalizes the domain to the unit hypercube. Ax also leverages several outcome transforms such as `StandardizeY`, which standardizes all metrics to have mean zero and variance one, and `BilogY`, which applies the bilog transform from Eriksson and Poloczek (2021) to outcome constraint metrics in order to magnify the region around the constraint boundary and improve performance on constrained problems.

**Acquisition**. The `Acquisition` class is responsible for constructing the BoTorch acquisition class and the necessary input arguments, as well as optimizing it to generate candidates. If the acquisition function class is not specified, it will pick between qLogNEI and qLogNEHVI (Ament et al., 2023) based on the number of objectives in the optimization config. Where possible, BoTorch leverages gradient-based methods to optimize the acquisition functions. To support continuous, discrete, and mixed search spaces, `Acquisition` dispatches to an appropriate optimizer from BoTorch depending on the features in the (post-transform) search space. This includes, using L-BFGS-B based optimizer for continuous parameters, local search or enumeration for discrete parameters, and an optimizer that alternates between discrete and continuous steps for mixed search spaces (Wan et al., 2021).

```
GeneratorSpec(
    model_enum=Generators.BOTORCH_MODULAR,
    model_kwargs={
        # Select between two models: An additive mixture of relatively strong SAAS priors
        # with learnable input warping and a relatively vanilla GP with a Matérn kernel.
        "surrogate_spec": SurrogateSpec(
            model_configs=[
                ModelConfig(
                    botorch_model_class=AdditiveMapSaasSingleTaskGP, input_transform_classes=[Warp],
                ),
                ModelConfig(
                    botorch_model_class=SingleTaskGP, covar_module_class=MaternKernel, covar_module_options={"nu": 2.5}
                ),
            ]
        ),
        # Negative integrated posterior variance as acquisition function for active learning.
        "botorch_acqf_class": qNegIntegratedPosteriorVariance,
    },
)
```

Code Example 3: Implementation of parallel active learning (negative integrated posterior variance) with a sparsity-inducing GP prior and input warping using a custom BoTorch generator in Ax. Specifying multiple `ModelConfig`s results in a "model selection" procedure picking the "best" model from the list, where "best" is defined as the model achieving the highest rank correlation based on leave-one-out cross validation (if desired, this can be separately configured further).

## C.2 Handling Outcome Constraints

In many applications, user wants to impose constraints on one of more outcomes. For instance, optimizing an ML model for capacity efficiency may mean performing tuning architectural parameters to increase throughput while not or only minimally regressing model quality. Ax allows incorporating such constraints in both single- and multi-objective settings. By default, Ax uses the `qLog(Noisy)ExpectedImprovement` or `qLog(Noisy)ExpectedHypervolumeImprovement` acquisition functions.

## D  Details on Diagnostics and Visualizations

Beyond algorithms and orchestration capabilities, Ax provides a number of diagnostic tools and visualizations that help the user understand various aspects of the underlying problem and the optimization. Especially in a human-in-the-loop context, this can be highly valuable for refining the problem statement and improving optimization outcomes.

The example visualizations in Figures 14 - 18 in this section are based on a single run on the LCBench (Zimmer et al., 2021) benchmark surrogate for the Fashion-MNIST data set (available under the Apache 2.0 license). In this problem, the goal is to optimize the accuracy of a fully-connected neural network over a 7-dimensional search space consisting of architecture parameters (the number of layers), optimizer parameters (learning rate, momentum, weight decay, batch size).

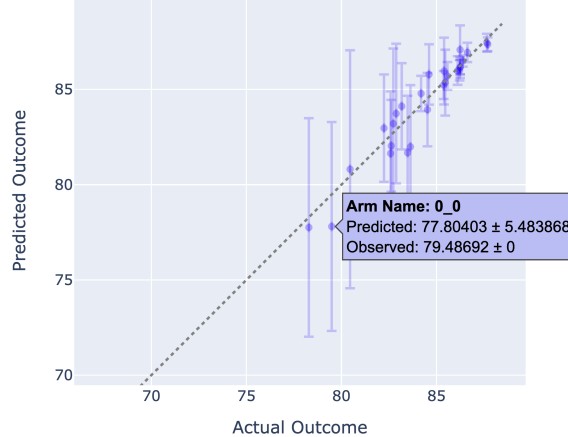

Figure 14: Leave-one-out cross-validation of the surrogate GP model used in Ax's default generation strategies. The plot shows posterior predictions for the ML model accuracy in terms of mean and 95% confidence intervals for each observation. In this example, we can see that the GP surrogate has good predictive performance and that the posterior variance is lower towards higher values, where the Bayesian Optimization algorithm has explored more configurations. The plot is interactive and users can get details on individual predictions by hovering over them with their cursor.

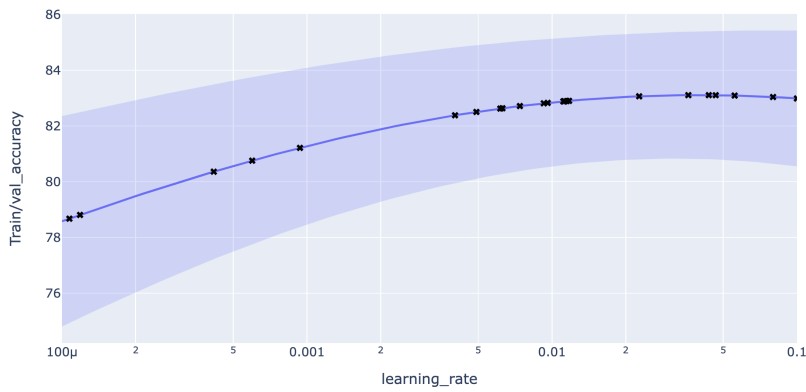

Figure 15: 1D Slice plot of model predictions of the ML model's accuracy as a function of the learning rate (other parameters fixed at reference). The plot shows predicted mean and 95% confidence interval as a function of the learning rate, as well as the observed values (black crosses).

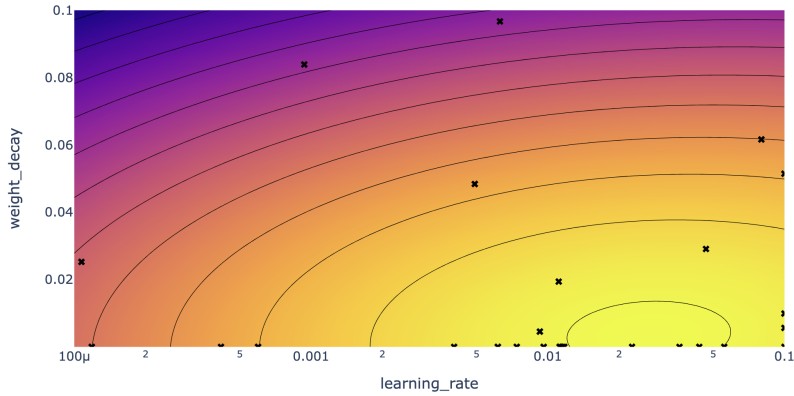

Figure 16: Contour plot visualization of the response surface of the negative validation loss as a function of the learning rate and weight decay parameters (brighter colors correspond to higher values). The behavior of how the accuracy depends on the learning rate from Figure 15 is also represented in this visualization.

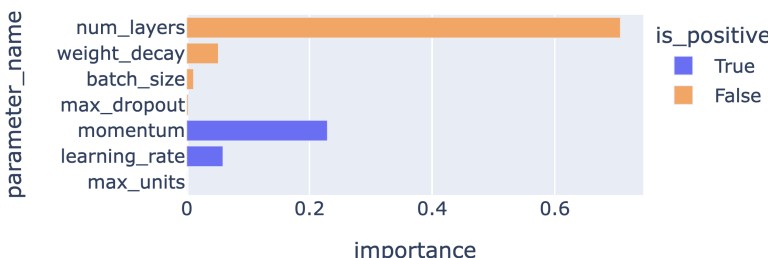

Figure 17: Global parameter sensitivity analysis plot based on Sobol indices (Sobol', 2001). Intuitively, the larger the sensitivity value, the more a particular parameter affects the outcome – "is_positive" denotes whether the effect is positive (True) or negative (False) as the parameter value increases. We have found that in practice our users highly value this and similar diagnostics.

## E   Details on Applications at Meta

### E.1   Optimizing Generative AI Models Using Human Preferential Feedback with Ax

One application of Ax at Meta is to optimize an auto-dubbing generative AI model, which translates and dubs audio into a different language while synchronizing the speaker's lip movements and facial expressions to match the new language (Meta, 2024b). While humans are quite good at assessing which version of a generated video appears most natural based on pair-wise comparisons, defining a "naturalness score" is extremely hard. By representing pairwise comparison data and leveraging state-of-the-art preference learning algorithms (Lin et al., 2022; Astudillo et al., 2023) through Ax, we have been able to substantially improve the quality of automatic dubbing and lip syncing AI translation tools as illustrated by Figure 19. In this setup, Ax generates a batch of configurations for which to train the AI model, the outputs of which on a set of evaluation videos are then compared pair-wise by humans, who indicate which version appears more natural to them.

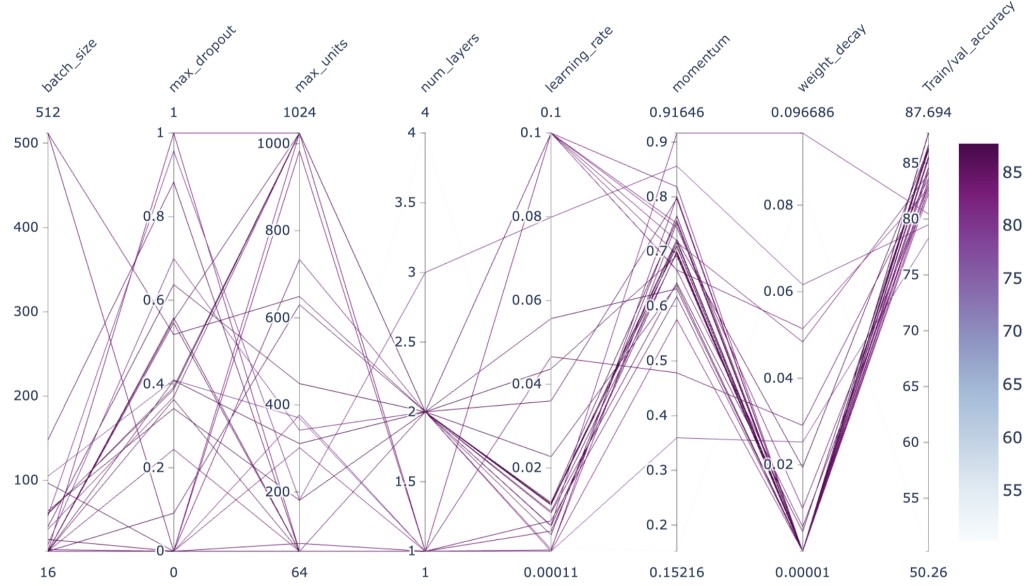

Figure 18: Parallel coordinate plot for the accuracy (indicated by color, darker colors corresponding to higher accuracy. The visualization presents more information about other parameters than the slice or contour plots, but may not be as easily interpretable.

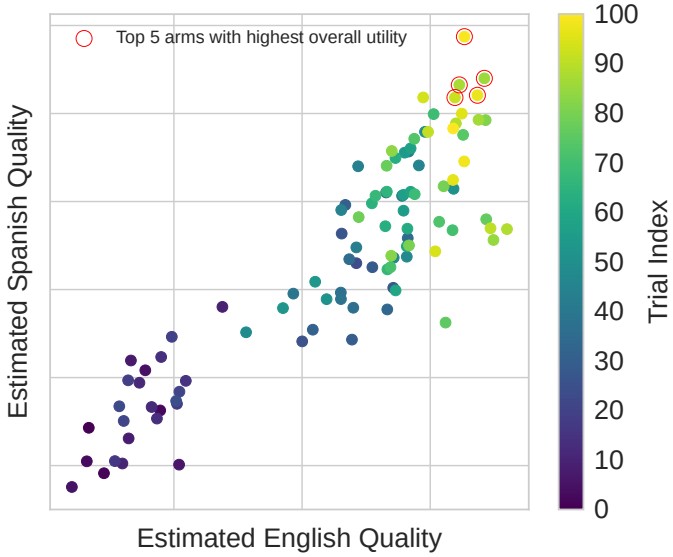

Figure 19: Estimated Pareto front of viewer preference on dubbed English and Spanish videos, respectively. Scales are omitted intentionally as the magnitude of the inferred video quality scores is on a relative scale. As we iterate, Ax improves the neural-dubbing model's performance by pushing the Pareto front outward without compromising the qualities of generated English and Spanish videos.

