# OpenReview forum: "Ax: A Platform for Adaptive Experimentation"
_automl.cc/AutoML/2025/ABCD_Track — AutoML 2025 ABCD Track_

### Official Review · Reviewer_ewhf · 2025-04-22

**Comments To Authors:**

### Summary
The authors introduce a black-box optimization library - Ax. It supports more features like parameter constraints or outcome constraints (e.g. model size in MB) than other available libraries (SyneTune, Optuna, Vizier, SMAC3, HEBO). At the same time, it performs better or similar to these benchmarks on many black-box optimization tasks - although SMAC3 and HEBO still have an edge on BBOB. The library is actively used in Meta, and it has a large user base as demonstrated per github stars and other metrics. The paper is overall very well written.

The paper structure is as follows - introduction and related work in Sections 1 and 2, main features in Section 3., comprehensive overview of API and workflows in Section 4., and experiments on black-box optimization problems in Section 5.

### Strengths
The paper structure has exceptional quality - it is easy to understand the main features, and advantages compared to other libraries, even for a person not directly working on black-box optimization (i.e. users). The API overview and code examples are a great summary of how to use the library. The experiments are well presented with additional details in the appendix

### Weaknesses
It was not clear to me whether online parameter tuning (via A/B testing) was available to non-Meta users or not. Also, it is not documented in detail how to replace the default BO with a different search algorithm (although the authors state that this feature is supported).

### Decision
The presented library is a nice addition to the HPO software ecosystem. The paper itself is a prototype for how library papers should be written. My rating is strong accept

**Review Confidence:**

3

**Review Rating:**

9

---

### Review · Reproducibility_Reviewer_BJQ7 · 2025-04-25

**Comments To Authors:**

The submitted code accompanying the paper is well documented and I had an easy time following the installation instructions. The only (very minor) hurdle was installing `cocoex` as part of the requirements on a fresh machine that didn't have a cc or gcc compiler, which was easily fixed.

**Review Confidence:**

4

**Review Rating:**

8

---

### Official Review · Reviewer_t7Vy · 2025-05-02

**Comments To Authors:**

This paper discusses the library Ax for adaptive experimentation, which functions in part as a way to interface bayesian optimization to production settings.

As a library paper, there is an emphasis placed on discussing the usage of the library, both from a numbers / usecase perspective as well as from the viewpoint of the API structure. This showcases that the library seems to be robust and thus could serve as a useful tool in AutoML pipelines. By providing a feature-based comparison to a variety of other frameworks, there is sufficient context provided for its role in the broader ecosystem. However, from this overview I'm missing a sense of the limitations / features which are (currently) unsupported, but are present in at least some of these alternatives, as that would make for a more complete comparison (e.g. multi-fidelity optimization, which is mentioned as in progress in section 7, would be useful to list in the comparison).

One aspect which is not mentioned is whether there is any inter-operability between Ax and the other tools discussed. Since a set of these tools were used in the benchmarking experiment, it might be worthwhile mentioning whether connecting these parts to the same problems was a manual effort, or whether connections (e.g. between search-space representations) are more broadly available.

I'm not sure about the evaluation procedure used for the noisy benchmarks. If I'm interpreting section 6.1 correctly, you use the best-so-far noiseless function value, but this would (potentially) differ from the function value the algorithm would actually return at that point. In 4.3 it is mentioned that a best point can be extracted at any time, which I assume can't correspond to this ideal noiseless value (since it might be unknown, maybe this would be useful to touch on briefly), so it would probably be more fair to use this point to determine anytime function values.

In Section 6.1, there is no mention of the number of evaluations used, which is often the key parameter to determine which method is the most suitable. While this information is later provided in the appendix, I feel it is critical to the interpretation of the experimental data, and should thus be included in the main paper. Since the total budget is either 50 or 100, this likely indicates that very exploitative methods would have some advantages, which is important context to have. Additionally, looking at the anytime performance, it seems that there could be differences in e.g. the fraction of the budget which is used in a DoE stage (looking at BBOB for HEBO and SMAC). While the general idea behind the used baselines is explained, it would be useful to also indicate any key parameter settings, or mention explicitly that the default values were used.

Some other minor questions / comments:
- I'm not quite sure what is meant by 'outcome constraints' in single-objective settings
- Section A.1 Generic algorithm -> Is this supposed to be genetic algorithm?
- What does 'randomly shifted versions of the BBOB problems' mean? BBOB already includes an instance generation procedure, does this refer to that or do you perform an additional shift?
- Why were ZDT1-3 and DTLZ2 chosen specifically from their broader benchmark suites?
- In figure 8, why does the Async group stop at ~20 evaluations? All problems in this set are listed as having 50 trials in table 3.

Overall, I think this paper provides a useful contribution to the ABCD track as a library paper introducing an open-source tool which can be useful for a variety of AutoML tasks, and thus I recommend its acceptance.

**Review Confidence:**

3

**Review Rating:**

8

---

### Meta-Review · Area_Chair_8ohV · 2025-05-08

**Recommendation:** Accept
**Confidence:** 5

**Metareview:**

The paper presents a platform for adaptive experimentation, called Ax, designed to solve black-box optimization problems such as hyperparameter optimization.
Reviewer ewhf highlighted the well-thought-out structure of the paper, and reviewer t7Vy appreciated its emphasis on the practical usage of the library.
While some minor issues were noted regarding specific details, all reviewers recommended acceptance.
I agree with this assessment and believe the paper is a strong fit for the ABCD track.